# Differentiating without Partial Evaluation

## Abstract

In the physical sciences, the gradient of a model is often simplified into a compact form ideal for a given context to be interpretable and more efficient; in fact, sometimes the efficiency of evaluation can be improved by an asymptotic factor due to symmetries. To learn interpretable surrogate models that accelerate physics simulations, a differentiation system capable of compact and unevaluated gradient expressions is highly desirable. However, standard symbolic and algorithmic differentiation both start by partially evaluating the model. After this points, the gradients irreversibly become blackboxes with potentially obscure performance ceilings. Based on the observation that composition is one of two combinators that form a complete basis with captures, we compliment the chain rule with a second rule that enables differentiation without any form of evaluation. Using a prototype implementation, we obtain compact gradient expressions for an MLP and a common physics model that, historically, resisted algorithmic differentiation. Lastly, we discuss the theoretical and practical limitations of our approach.

## 1 Introduction

What is the gradient of a composition $w \mapsto f(g(w))$? The obvious answer is the chain rule $w \mapsto g'(w) \cdot f'(g(w))$, which is a powerful tool because it allows us to differentiate by rearranging instead of evaluating $f$ or $g$ to specific functions. For example, using the chain rule, we can differentiate $x \mapsto (x+1)^n$ as $x \mapsto n(x+1)^{n-1}$ without a binomial expansion, which would lead to a very large differentiation problem. Let us now consider $w \mapsto f(w)(g)$, which is a slightly modification that leads to a great deal of hardship because it is not a composition. The simplest answer to this is tracing Baydin et al. (2015); Elliott (2018); Griewank and Walther (2008), which is a form of partial evaluation Innes (2018) that runs the function with specific values of $f$, $g$, and $w$ while recording all primitive operations applied to $w$ as a computation graph to obtain a composition. For example, if we trace the computation with $f = x \mapsto v \mapsto xv$, $g = 2$, and $w = 1$, we find that $w$ is multiplied by 2, so it suffices to differentiate $w \mapsto 2w$. To summarize, the chain rule enables differentiating a function without evaluating it, as long as the function is a composition. However, when this assumption breaks, one typically needs to **partially evaluate** the function numerically or symbolically until it is a composition. In source transform system, this problem can be addressed through functional approaches Vytiniotis et al. (2019); Elsman et al. (2022); Pearlmutter and Siskind (2008); Ehrhard and Regnier (2003), although expressing scientific applications as functional programs remains difficult.

This is not a fictitious problem. In particular, we will see that tensor operations, which is the building blocks for many physics models, also fall into this category. For differentiating tensor operations, it is essential that we retain a symbolic form for two reasons. First, tensors in Physics have symmetries, which can be used to simplify the gradient. The simplification often leads to a performance gain by an integer or even an asymptotic factor. Second, the gradient needs to be interpretable because it usually represents the physical law of the theory, which may contain as much insight as its numerical solution. These two requirements make algorithmic differentiation less attractive. Specifically, tracing based systems such as PyTorch Paszke et al. (2017; 2019) and JAX Bradbury et al. (2018a) partially evaluates the models into a computation graph which represents the gradient. This is clearly amenable to neither interpretation nor symbolic manipulation. The other alternative is source transformation such as Enzyme Moses and Churavy (2020); Moses

et al. (2021; 2022) and Tapenade Hascoet and Pascual (2013), which either requires the code to be written in low-level procedural primitives or compiles the code to intermediate representations, thus removing the abstraction $w \mapsto f(w)(g)$. The last common option is symbolic differentiation such as SymPy Meurer et al. (2017) and Mathematica, which appears to fit our requirements, but differentiating functions of the type $w \mapsto f(w)(g)$ still requires partial symbolic evaluation, which is susceptible to expression swell Baydin et al. (2015) and is not suitable for general problems such as ML models, which we also need.

In this paper, we show that adding a second differentiation rule in addition to the chain rule makes it possible to delay all evaluation until after the differentiation. This enables a symbolic differentiation system that avoids common cases of expression swell, which is not too different from a source transformation system enriched with symbolic capabilities. As the result, we obtain gradients for tensor operations that can be interpreted and simplified using symmetries. Additionally, because the output of the differentiation is also tensor operations in symbolic form, it can be fed into tensor operation engines, thus resolving difficulties in the gradient code such as differentiability constraints Bradbury et al. (2018a;b); Lukas Devos and contributors (2023) and handling tensor contraction order cuTENSOR; Georganas et al. (2021); Fishman et al. (2020); Hirata (2003); Abbott et al. (2023).

Specifically, similar to how the chain rule avoids some of the evaluation when differentiating, we find a second rule that avoids the rest of the evaluation, thus narrowing the gap between symbolic and algorithmic differentiation. The **key insight** that enables this work is that composition is one of two "bolts and nuts" of mathematical functions, which can be used to assemble an arbitrary function from a few univariate primitives. Formally, the two components are known as the **B** (composition) and **C** combinators in combinatory logic Schönfinkel (1924), and their differentiation rules are straightforward to derive. Notably, we also allow for captures in the composed function, so the **S** combinator may be a more appropriate description than **B**. These combinators has been primarily used to study computability Curry et al. (1958). In practice, it has been used for building parsers Fokker (1995); Leijen and Meijer (2001), reasoning about data updates Foster et al. (2007), automatic parallelization Lafont (1997), as well as extending AD frameworks Lin (2023) as high-level primitives.

To reiterate, our main message is a **qualitative claim** that a second differentiation rule in addition to the chain rule enables differentiation without evaluation, which leads to gradient expressions for tensor operations that can be simplified using symmetries and interpreted with physical meaning. The result is demonstrated via a prototype implementation that produces compact gradient expressions for a representative set of examples. It is worthwhile to clarify that this paper is **neither** suggesting a new high-level or low-level differentiation framework, **nor** does it claim to achieve quantitatively better efficiency for any class of problems. As an outline, we start the paper with some notation and background of AD. We then present the theoretical model and illustrate how combinators help us bypass partial evaluation in either numerical or symbolic form. We provide a prototype implementation with MLP, Hartree-Fock (HF) Thijssen (2012); Slater (1951), and conjugate gradient Hestenes and Stiefel (1952); Trefethen (2022) as examples, which typically require partial evaluation of some form to differentiate. Lastly, we discuss the class of problems that we are limited to and engineering complications that we face.

## 2 NOTATION

We use anonymous functions (lambdas) extensively so that we can write functions as values such as $x \mapsto x + 1$ instead of definitions $f(x) = x + 1$. This makes it easy to think of functions as inputs and outputs of other functions. Moreover, we adopt the anonymous notation for expressing tensors and treat them as maps from integer indices to their corresponding tensor elements. Invoking a tensor as a function is the same as indexing. For example, $v(i)$ and $v_i$ are equivalent. Similarly a tensor can be constructed as a function. For example, $i \mapsto 2v(i)$ is the same as $2v$.

We will also make extensive use of delta functions, including the Kronecker delta function

$$\delta(i, j, k) = \begin{cases} k & \text{if } i = j \\ 0 & \text{otherwise} \end{cases}, \tag{1}$$

The main identity regarding the delta function that we will use is the contraction theorem

**Theorem 1** *Delta contraction theorem: the contraction of the delta function with a function f results in a substitution of the argument of f.*

$$\Sigma(i \mapsto \delta(i, j, f(i))) = f(j). \tag{2}$$

*This holds when $i, j$ are integers, real and complex numbers, tensors, as well as continuous functions.*

This theorem is well-known and easily checked when $i, j$ are integers, which means that $\delta$ is a Kronecker delta and $\Sigma$ is a simple sum. The more general cases are less intuitive but is algebraically derived in the appendix. For the purpos of this paper, it suffices to understand the integer case and accept that an algebraic generalization is possible.

## 3 BACKGROUND

To put our theory in the context of AD, we briefly introduce the theory for reverse mode AD, which is based on the composition and the chain rule. In a AD system based on tracing, the chain rule is typically presented in multivariate calculus with Jacobian products. Given a composite function $g(x) = f_1(f_2(x))$, where $f_1 \in \mathbb{R}^N \to \mathbb{R}$ and $f_2 \in \mathbb{R}^M \to \mathbb{R}^N$, the gradient of $g$ is

$$\nabla g(x)^T = \mathcal{J}g(x)^T = \mathcal{J}f_2(x)^T \cdot \mathcal{J}f_1(f_2(x))^T. \tag{3}$$

Computing and multiplying the full Jacobian matrices can be inefficient in practice if the Jacobian is sparse. For example, if $f_2$ is an element-wise map, then $\mathcal{J}f_2(x)$ is diagonal.

Instead, one can encode the Jacobian through its action on some vector $k$ using pullbacks

$$\mathcal{P}f(x, k) = \mathcal{J}f(x)^T \cdot k = i \mapsto \sum_j k_j \partial f(x)_j / \partial x_i, \tag{4}$$

where we have used anonymous notation and denoted a vector by describing its $i^{\text{th}}$ element. As a simple example, the pullback of multiplication by a scalar is

$$\mathcal{P}(x \mapsto vx) = (x, k) \mapsto k \frac{\partial(vx)}{\partial x} = kv. \tag{5}$$

Using pullbacks, the Jacobian chain rule can be written in terms of its actions

$$\mathcal{P}g(x, k) = \mathcal{J}f_2(x)^T \cdot (\mathcal{J}f_1(f_2(x))^T \cdot k) = \mathcal{P}f_2(x, \mathcal{P}f_1(f_2(x), k)), \tag{6}$$

$$\nabla g(x) = \mathcal{P}g(x, 1) = \mathcal{P}f_2(x, \mathcal{P}f_1(f_2(x), 1)). \tag{7}$$

In reverse mode AD, the forward pass is essentially the evaluation of $f_2$ at $x$ and the reverse pass is the evaluation of $\mathcal{P}f_1$ and $\mathcal{P}f_2$ with their respective arguments.

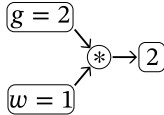

Figure 1: The computation graph traced from computing $f(w)(g)$. The graph represents $w \mapsto 2w$.

When differentiating a function that is not a composition such as $w \mapsto f(w)(g)$, the standard strategy is to convert it to a composition via partial evaluation. One can do this numerically and record all primitive operations [1]. This recording can be represented as a graph where nodes are calls to primitives and edges are data. For example, running $f(w)(g)$ with $w = 1$, $g = 2$, and $f = x \mapsto v \mapsto x * v$ yields a single nodes graph shown in fig. 1. In the end, the path from $w$ to the sink is the composition of functions applied to $w$. Alternatively, one can also do symbolic partial evaluation, which means substituting $x$ with $w$ and $v$ with $g$ to find $w \mapsto wg$.

---

[1]The tracing can be controlled to save only the necessary states.

## 4 THEORY

### 4.1 DERIVATIVES OF COMBINATORS

In section 1, we discussed the complications of partial evaluation and we claimed that the use of combinators can help us avoid it. Towards that end, we introduce the combinators and their differentiation rules. The definitions of $\mathbf{B}$ is

$$\mathbf{B} = f \mapsto (g \mapsto (x \mapsto f(g(x)))), \quad x \in X, g \in (X \to Y), f \in (Y \to Z), \tag{8}$$

Intuitively, $\mathbf{B}$ takes two functions $f$ and $g$, composes them, and applies the result to $x$. $f, g, x$ can be in any space as long as they are consistent, meaning that the $x$ is in the same space as the input to $g$ and the output of $g$ is in the same space as the input of $f$. This constraints is specified by the generic spaces $X, Y, Z$. Similarly, $\mathbf{C}$ is defined as

$$\mathbf{C} = f \mapsto (x \mapsto (y \mapsto f(y)(x))), \quad x \in X, y \in Y, f \in (Y \to (X \to Z)), \tag{9}$$

which takes $f$ as the input and return a function that swaps the first two arguments. For example, $\mathbf{C}(x \mapsto v \mapsto k) = v \mapsto x \mapsto k$. $f, x$ and $y$ can be in any spaces as long as they are consistent.

Using these combinators, a function can be decomposed into a small set of primitives through a process called abstraction elimination Sørensen (2006). For example, $w \mapsto f(w)(g)$ can be written as $w \mapsto B(f(w))(I)(g)$. Since we are differentiating with respect to $w$ instead of $g$, the chain rule needs to be modified

**Theorem 2** *The differentiation rule for the $\mathbf{B}$ combinator with captures is*

$$\begin{aligned} \mathcal{P}\left(x \mapsto \mathbf{B}(f)(g)(x)\right) = (x, k) \mapsto \\ \mathcal{P}\left(g\right)\left(x, \mathcal{P}\left(f\right)(g(x), k)\right) + \mathcal{P}\left(x \mapsto f\right)\left(x, i \mapsto \delta(g(x), i, k)\right), \end{aligned} \tag{10}$$

*where the first term is the chain rule and the second term is new. No evaluation is involved in the rule because $f, g$ are not specified. Equivalently, one can model the captures with the $\mathbf{S}$ combinator, which is defined as*

$$\mathbf{S} = h \mapsto (g \mapsto (x \mapsto h(x)(g(x)))). \tag{11}$$

*The $\mathbf{B}$-rule with captures can be rephrased as the $\mathbf{S}$-rule*

$$\begin{aligned} \mathcal{P}\left(x \mapsto \mathbf{S}(h)(g)(x)\right) = (x, k) \mapsto \\ \mathcal{P}\left(g\right)\left(x, \mathcal{P}\left(h(x)\right)(g(x), k)\right) + \mathcal{P}\left(h\right)\left(x, i \mapsto \delta(g(x), i, k)\right). \end{aligned} \tag{12}$$

Similarly, we have

**Theorem 3** *The differentiation rule for $\mathbf{C}$*

$$\mathcal{P}\left(\mathbf{C}(g)\right) = (x, k) \mapsto \Sigma(b \mapsto \mathcal{P}\left(g(b)\right)(x, k(b))), \tag{13}$$

*which also involves no evaluation because $g$ is not specified.*

These rules can be derived by plugging the definitions of the combinators into the definition of the pullback (see section 1 in the appendix), and the $\mathbf{B}$-rule is known in lambda calculus form Ehrhard and Regnier (2003). Importantly, the application of either rule only rearranges existing symbols without evaluating any of them to specific functions or numbers.

### 4.2 DELAYING EVALUATION

To explicitly illustrate how the differentiation rules eq. (10) and eq. (13) delay evaluation, let's revisit the differentiation of $w \mapsto f(w)(g)$ using the $\mathbf{B}$-rule

$$\mathcal{P}(w \mapsto B(f(w))(I)(g))(w, 1) = \underbrace{\mathcal{P}(w \mapsto I(g))(\ldots)}_{0} + \mathcal{P}(f)(w, i \mapsto \delta(g, i, 1)), \tag{14}$$

which is the analytic solution to our original problem in terms of $f$, $g$, $w$ and their pullbacks. Notice how the manipulation is restricted to moving symbols and neither $f$ nor $g$ is evaluated to specific functions. If we separately differentiate $f = x \mapsto v \mapsto xv$ using the **C**-rule, we find

$$\mathcal{P}(C(v \mapsto x \mapsto xv))(y,k) = \Sigma(v \mapsto \mathcal{P}(x \mapsto vx)(y,k(v))) \tag{15}$$

$$= \Sigma(v \mapsto ((x,k) \mapsto vk)(y,k(v))). \tag{16}$$

Similarly, the application of **C** in eq. (15) has only rearranged the symbols without evaluating any function. The last step eq. (16) is simply looking up the pullback of a primitive (multiplication by a constant). Combining eq. (14) and eq. (16) gives us $\Sigma_v v\delta(g,v,1) = g$, which implements a symbolic evaluation using eq. (2), but only after the differentiation. In the last step, we have invoked a general form of eq. (2), because $v$ and $g$ can be matrices or functions, which makes the algebraic generalization necessary.

### 4.3 Tensor Operations

As a second illustration, we differentiate a basic tensor operation $f \mapsto \Sigma_j f(j)$, which can be written as $f \mapsto \Sigma(j \mapsto \mathbf{B}(f)(I)(j))$ or $\Sigma(j \mapsto \mathbf{S}(I)(v \mapsto j))$ and the **B**-rule reads

$$\mathcal{P}(f \mapsto \Sigma(j \mapsto f(j))(y,1)) = \Sigma(i \mapsto \mathcal{P}(f \mapsto f)(y, i \mapsto \delta(i,j,1))) = i \mapsto 1, \tag{17}$$

where we have looked up the pullback of the identity primitive $f \mapsto f$, which is $(f,k) \mapsto k$. The result states that evaluating the pullback of summation is an array of ones, and the compiler should be able to map the final expression to a single call to `memset`. Generally, both the model and gradient expressions are to be compiled to either IO optimized primitives or for loops around mutations. This approach is to be contrasted against symbolically expanding $f$ into $[f_1, f_2, \ldots]$ or implementing the expression to low-level primitives `sum += f[i]`, which are symbolic evaluations of $f$ or $\sum$ into lower-level forms. The summation is treated as a primitive for brevity, but its differentiation rule will be derived from the two rules in the appendix. This derivation may be reminiscent of the more familiar variational differentiation Gelfand (2000)

$$\partial \Sigma_j f_j / \partial f_i = \Sigma_j \delta_{ij} = 1. \tag{18}$$

A more sophisticated tensor operation that is practically relevant will be found in section 5.2, but the principles are the same.

### 4.4 Nonlinearity (Fanout)

Nonlinear functions requires fanout, which is handled by the second part of the **B**-rule. For example, the binary product rule can be derived from the unary product rule as (keeping the two operants the same for simplicity).

$$\begin{aligned}
&\mathcal{P}(x \mapsto x \cdot x)(x,k) \\
=&\mathcal{P}(x \mapsto (v \mapsto x \cdot v)(x))(x,k) \\
=&\mathcal{P}(x \mapsto x)(x, \mathcal{P}(v \mapsto x \cdot v)(x,k)) + \mathcal{P}(x \mapsto (v \mapsto x \cdot v))(x, i \mapsto \delta(i,x,k)) \\
=&\mathcal{P}(v \mapsto x \cdot v)(x,k) + \sum_v \mathcal{P}(x \mapsto x \cdot v)(x, \delta(v,x,k)) \\
=&xk + \sum_v v\delta(v,x,k) = 2xk.
\end{aligned} \tag{19}$$

This approach is to be contrasted in hardcoding the product rule.

### 4.5 Ordered Iterations

In the fanout example, we have modeled a binary product as a unary product that captures $x$ applied to $x$. This can be inductively generalized to an ordered sequence of functions with captures.

$$\bigwedge f = f(N) \circ f(N-1) \circ \ldots \circ f(1), \tag{20}$$

where the size parameter $N$ is assumed to be encoded in the domain of $f$. For convenience, we will use $\bigwedge_i f(i)$ interchangeably with $\bigwedge f$. Alternatively, following the **S** combinator, we can define

$$\Gamma h = x \mapsto h(x)(N) \circ h(x)(N-1) \circ \ldots \circ h(x)(1). \tag{21}$$

For example, a monomial $x \mapsto x^N$ can be defined as

$$x \mapsto \left( \bigwedge (i \mapsto v \mapsto x \cdot v) \right)(1). \tag{22}$$

The differentiation rule for $\bigwedge$ is inductively derived form the two basic rules in the appendix, where we also show that the rule for $\bigwedge$ can be viewed as an abstraction over the typical prescription of differentiation by source transform as shown on page 125-127 in Griewank and Walther (2008). This rule in itself is not able to differentiate a model with respect to its parameters such as the MLP, but the benefit of an abstract rule is that it trivially extends to parametric models by composing with the other two rules as demonstrated in section 5.1. The rule for $\bigwedge$ does not cover fixed point iterations, which will be discussed separately in the appendix. Unordered iteration can be differentiated this way, but they should be handled as tensor operations to avoid imposing unnecessary order.

### 4.6 MUTATIONS

It is possible to differentiate through mutations under our framework, but we choose to avoid supporting it in favor of directly writing and differentiating tensor operations. The tensor operations are to be mapped to IO optimized primitives or for loops around mutations by the compiler. Nevertheless, we will illustrate how to differentiate mutations for illustration. First, we bring mutations into a functional form that is confluent to the original statement. For example, `x[i] += 1` is confluent to $x = (j) \mapsto x(j) + \delta(i, j, 1)$, which maps $x$ to an updated new vector. The functional form can be differentiated first by applying the C rule

$$\mathcal{P}(x \mapsto j \mapsto x(j) + \delta(i,j,1))(x,k) = \sum_j \mathcal{P}(x \mapsto x(j) + \delta(i,j,1))(x, k(j))$$

Then we apply the B rule to get

$$\sum_j i \mapsto \delta(i,j,k(j)) = i \mapsto k(i) = k.$$

Therefore, we have concluded that incrementing an array element by a constant merely adds an identity map in the backward phase, which can be optimized away by the compiler. Other types of mutations can be derived analogously.

## 5 EXAMPLES

Once partial evaluation is circumvented, the line separating AD and symbolic differentiation starts to blur. This is because the two methods differ largely in the compromises they make to accommodate partial evaluation. AD is efficient but blackbox, whereas symbolic differentiation is transparent but suffers expression swell.

With the two combinators at hand, one no longer needs to accept such a nuanced trade-off due to the use of partial evaluation and can simultaneously enjoy both efficiency and transparency. Concretely, this means we can differentiate code and obtain an gradient expression resembling the handwritten gradient for problems that typically require building computation graphs or differentiating low-level code. For demonstration, we now showcase differentiating a MLP and the HF energy, which are classic examples where numerical or symbolic partial evaluation is used for differentiation.

Our proof-of-concept system is implemented as a domain specific functional programming language within Julia, the source code of which will be provided along with all examples. The language supports most necessary ingredients of functional programming such as closures, conditionals, and let statements. The main missing piece is recursion, which has not been the appropriate iteration facility for scientific applications. Instead, unordered iteration is supported via tensor expressions, which can be mapped to optimized tensor engines. Future support for ordered iteration and fixed point iteration will be discussed in section 2 of the appendix.

## 5.1 MLP

The most representative problem of backpropagation is the MLP. We show that we can generate a unevaluated and readable gradient. As shown in listing 1, we have an MLP whose weights are `w_1`, `w_2`, and `w_3`, which are of type `RM` (real matrices) and `RV` (real vectors). The first layer is `(x::RV) -> mvp(w_1, x)`, which is a function that multiplies its input by `w_1`. The output of this layer is piped (`|>`) to the second layer, which is element-wise nonlinear activation. Notice that `ReLU` is defined in the outer scope as an unknown of type `RF` (functions from real to real). After defining the model, we apply it to the sample `batch`. Lastly, we surround the function that we like to differentiate with the keyword `pullback`.

Listing 1: Multilayer perceptron

```
mvp = (A::RM, x::RV) -> (i::N) -> sum(j, A(i, j) * x(j))
vip = (x::RV, y::RV) -> sum(i, x(i) * y(i))
(batch::RV, Relu::RF) ->
    pullback((w_1::RM, w_2::RM, w::RV) ->      # define weights
        (((x::RV) -> mvp(w_1, x)) |>           # linear layer 1
        ((x::RV) -> (i::N) -> Relu(x(i))) |>   # nonlinear activation 1
        ((x::RV) -> mvp(w_2, x)) |>            # linear layer 2
        ((x::RV) -> (i::N) -> Relu(x(i))) |>   # nonlinear activation 2
        ((x::RV) -> vip(x, w)))(               # prediction
            batch                              # apply to sample
        ))
```

Differentiating this code yields the gradient shown in listing 2, which describes backpropagation for this specific MLP as concisely as one would hope for. Notice that the definition of `mvp` and `vip` do not appear in the gradient because these functions have never been evaluated numerically or symbolically. The pullback of `ReLU` appears in the backward pass even though the program is oblivious of what it is.

Listing 2: MLP gradient

```
let
    mvp = (A, x) -> (i) -> sum((j), x(j)*A(i, j))
    vip = (x, y) -> sum((i), x(i)*y(i))
    (batch, Relu) -> (w_1, w_2, w_3) -> let
        _y = mvp(w_1, batch)                        # |
        _y_1 = (i) -> Relu(_y(i))                   # |
        _y_2 = mvp(w_2, _y_1)                       # |
        _y_3 = (i) -> Relu(_y_2(i))                 # | forward
        _y_4 = vip(_y_3, w_3)                       # v pass
        _l = P((_z) -> vip(_z, w_3))(_y_3, 1)       # |
        _l_1 = (_a) -> P(Relu)(_y_2(_a), _l(_a))    # |
        _l_2 = P((_z_1) -> mvp(w_2, _z_1))(_y_1, _l_1) # | backward
        _l_3 = (_a) -> P(Relu)(_y(_a), _l_2(_a))    # v pass
        tuple(P((_z) -> mvp(_z, batch))(w_1, _l_3), # |
            P((_z) -> mvp(_z, _y_1))(w_2, _l_1),    # |
            P((_z_2) -> vip(_y_3, _z_2))(w_3, 1))   # | gradient
    end
end
```

One may notice that our expression is suboptimal in memory usage compared to an optimized AD library, which accumulates the gradient instead of allocating memory for every sample. This is an example of how subtle performance questions in reverse mode differentiation become obvious once a compact gradient expression is available. Moreover, instead of building the accumulation into the differentiation library, we can leave it to the simplification stage, which can decide whether to accumulate based on the context. In principle, other performance optimization such as pipelining and recomputation Huang et al. (2018); Feng and Huang (2018) can also potentially be left to the simplification stage instead of extending an AD library itself.

## 5.2 Hartree Fock

As an example scientific application, we showcase taking the derivative of HF energy, which is representative of a rather different class of problems common in electronic structure theories Lin and Lu (2019); Martin (2004): tensor contractions. To differentiate tensor contractions, one can either trace out the low-level primitives, which is leads to very large graphs. One can also decompose it as a sequence of matrix operations in many different ways, but finding the way that minimizes memory and compute is hard. Most importantly, there are often duplicated terms that arise during differentiation due to symmetries, which must be leveraged to be comparable with hand-written gradients that are then implemented directly. Some physics terminology will be used in this example for an accurate presentation, but no physics background is needed to understand the message.

We consider the Coulomb energy term in the HF energy, which is a contraction between a fourth order complex tensor $J \in \mathbb{C}^{N \times N \times N \times N}$ and a complex matrix $C \in \mathbb{C}^{N \times N_e}$

$$J \mapsto C \mapsto \Sigma_{i,j,p,q,r,s} C^*_{p,i} C^*_{r,j} J_{p,q,r,s} C_{s,j} C_{q,i}. \tag{23}$$

One expects four terms in the derivative because the function is quartic, but all four terms are the same due to tensor symmetries. Thus, the gradient should be a single term, which also happens to be an intermediate of the Coulomb energy itself, so the gradient evaluation barely incurs an extra cost.

There are two symmetries of $J$ that enable the simplification: $J_{pqrs} = J^*_{qpsr}$ and $J_{pqrs} = J_{rspq}$, each of which is specified as a transform on $J$ as an invariant. For examples, consider the transform $f = J \mapsto (a,b,i,j) \mapsto J(i,j,a,b)$, it is easy to verify that $f(J) = J$ implies $J_{pqrs} = J_{rspq}$. The code for the symmetries and the Coulomb energy is shown in listing 3, where CM denotes complex matrices and (N, N, N, N) -> R states that $J$ is a map from four natural numbers to a complex number (i.e. fourth order complex tensor)

Listing 3: Hartree Fock

```
@space ERI begin
    type = (N, N, N, N) -> C
    symmetries = (J -> (j, i, b, a) -> J(i, j, a, b)',
                  J -> (a, b, i, j) -> J(i, j, a, b))
end

(J::ERI) -> pullback((C::CM) ->
    sum((i, j, p, q, r, s), C(p, i)' * C(q, i) * C(r, j)' * C(s, j) *
                            J(p, q, r, s)))
```

Without considering symmetries, the generated gradient consists of four distinct terms shown in listing 4. If we leverage symmetries, the compiler can detect that the four terms are in fact the same and combine them into one. This example is also an independent illustration of how combinators bypass partial evaluation, since no symbol has been evaluated symbolically or numerically.

Listing 4: HF gradient

```
# Without symmetry
 (J) -> (C) -> (_a, _a_1) -> (
 sum((i, p, q, r), J(p, q, r, _a) * C(p, i) * C(r, _a_1) * C(q, i)')+
 sum((j, p, r, s), J(p, _a, r, s) * C(p, _a_1) * C(r, j) * C(s, j)')+
 sum((j, q, r, s), J(_a, q, r, s) * C(s, j) * C(q, _a_1) * C(r, j)')+
 sum((i, p, q, s), J(p, q, _a, s) * C(s, _a_1) * C(q, i) * C(p, i)'))
# With symmetry
 (J) -> (C) -> (_a, _a_1) -> sum((i, p, q, r),
    J(q, p, _a, r) * C(p, i) * C(r, _a_1) * C(q, i)') * 4.0
```

## 5.3 Conjugate gradient

Lastly, we show an example where the gradient expression can be used for algorithmic insight with a conceptually simple and novel derivation of the conjugate gradient (CG)

algorithm Hestenes and Stiefel (1952); Trefethen (2022). The idea is to minimize $R = x \mapsto \frac{1}{2}x^T A x - b^T x$, whose stationary condition yields $Ax = b$. We consider a general momentum-like optimization step parametrized by the step sizes $x + \alpha(r + \beta p)$, where $x \in \mathbb{R}^N$ is the current iterate, $p \in \mathbb{R}^N$ is the previous step direction and $r \in \mathbb{R}^N$ is the current gradient. The residual as a function of the parameters after taking the gradient step is $(\alpha, \beta) \mapsto R(x + \alpha(r + \beta p))$, which we minimize with respect to $\alpha$ and $\beta$.

In listing 5, we first differentiate the residual without substituting the objective function $R$ to obtain an abstract theory that is generally applicable. Then we show that replacing $R$ with the quadratic form gives the CG coefficients.

Listing 5: Conjugate gradient

```
(A::Sym, r::RV, p::RV, b::RV, x::RV) -> begin
    R = (x::RV) ->
        sum((i, j), 0.5 * x(i) * A(i, j) * x(j)) - sum(i, x(i) * b(i))
    pullback((alpha::R, beta::R) ->
        R((i::N) -> x(i) + alpha * (r(i) + beta * p(i))))
end
```

The result of line 9 of listing 5 is shown in eq. (24), which gives a vector of two components. This shows that we can differentiate through unknown functions as a consequence of avoiding the partial evaluation.

$$
\begin{aligned}
(A, r, p, b, x) \mapsto &\text{ let}\\
R = &x \mapsto (-1.0 \cdot x^T \cdot b + 0.5 \cdot x^T \cdot A \cdot x)\\
(\alpha, \beta) \mapsto &(\nabla(R)((\alpha \cdot (\beta \cdot p + r) + x))^T \cdot (\beta \cdot p + r),\\
&\nabla(R)((\alpha \cdot (\beta \cdot p + r) + x))^T \cdot p \cdot \alpha)\\
&\text{end}
\end{aligned}
\tag{24}
$$

If we write $p_k = r + \beta p$, $\nabla R(\alpha p_k + x)^T \cdot p_k = 0$ has the interpretation that the gradient at the next iterate should be orthogonal to the current step direction. Combined with $\nabla R(\alpha p_k + x)^T \cdot p \cdot \alpha = 0$, we have a nonlinear system of two equations for $\alpha$ and $\beta$, the coefficients and thus the solutions of which depends on $R$.

Once we substitute the quadratic form for $R$ in line 11 of listing 5, the gradient reduces to

$$
\begin{aligned}
(A, r, p, b, x) \mapsto (\alpha, \beta) \mapsto &\\
(((\beta \cdot p + r)^T \cdot A \cdot (\alpha \cdot (\beta \cdot p + r) + x) - 1.0 \cdot (\beta \cdot p + r)^T \cdot b),&\\
\alpha \cdot (p^T \cdot A \cdot (\alpha \cdot (\beta \cdot p + r) + x) - 1.0 \cdot b^T \cdot p)).&
\end{aligned}
\tag{25}
$$

Using the fact that the gradient $r = Ax - b$ is orthogonal to the previous step direction $p$, the two nonlinear equations can be solved by hand to get

$$
\alpha = \frac{p_k^T \cdot (b - Ax)}{p_k^T A p_k} = -\frac{(r + \beta p)^T r}{p_k^T A p_k} = -\frac{r^T \cdot r}{p_k^T A p_k},
\tag{26}
$$

$$
\beta = \frac{(b^T - x^T \cdot A) \cdot p - \alpha r^T \cdot A \cdot p}{\alpha p^T A p} = \frac{r^T A p}{p^T A p},
\tag{27}
$$

which can be recognized as the parameters that produce the conjugate gradient method.

## 6 LIMITATIONS

### 6.1 THEORETICAL LIMITATIONS

The main limitation of our theory is that it requires stable dimensions, which means that the tensor dimensions cannot depend on the values in the tensor. For example, `filter` is not dimensionally stable because the length of the output depends not only on the length but also the values of its input. On the other hand, `map` is dimensionally stable because the length of the output is determined only by the length of its input.

The second theoretical limitation is not supporting mutations, although a substantial number of scientific applications are implemented as updating big tensors in a loop. In principle, mutations can be supported as shown in section 4.6, but allowing mutations encourages the users to use low level primitives to implement tensor operations, which make symbolic simplification impossible. Instead, we let the compiler implement the specified tensor operations as it sees fit either by offloading to IO optimized primitives or low-level mutating primitives.

### 6.2 Practical limitations

Aside from general software quality problems, we do not have a reliable implementation of fixed point iteration or the general sequential iteration, which we claim is possible in theory. These constructs are necessary for ODE constrained optimization problems and sensitivity analysis Fiacco and McCormick (1990); Gould et al. (2016), so the applicability of our approach to these problems still needs to be demonstrated in practice.

Tracing not only evaluates the values, but also the types of tensors, which is necessary information for differentiation. Since our differentiation happens entirely at compile time, we basically require a static type system. This does not integrate well with the dynamic type system in `Python` or `Julia`, and a separate or restricted type system is necessary.

## 7 Conclusion

Motivated by combinatory logic and scientific applications, we showed that introducing a second differentiation rule in addition to the chain rule allows us to avoid partial evaluation in either numerical or symbolic form for tensor operations and parametrized models. The result of the differentiation can be simplified and interpreted. Using a proof of concept functional programming language, we demonstrated that the theory can be implemented concretely and one can obtain readable expressions for gradients, even if the problem is not a composition and only partially known. We have also pointed out cases where our method does not work and issues that still need to be resolved in our implementation. Nevertheless, there appears to be a path to overcome partial evaluation and accelerate scientific endeavors.

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

OVERVIEW

In section 8, we show the derivation of the **B**-rule and the **C**-rule. In section 9, we extend our formalism to sequential iterations and fixed points. Lastly, in section 11, we explain our treatment of complex numbers.

## 8 DERIVATIVES OF COMBINATORS

### 8.1 DIFFERENTIATING **C**

Given a function $f : \mathbb{R}^M \mapsto \mathbb{R}^N$, recall that our definition of pullback is

$$\mathcal{P}(f) = (x, k) \mapsto \left( i \mapsto \Sigma \left( b \mapsto k(b) \frac{\partial f(x)(b)}{\partial x(i)} \right) \right). \tag{28}$$

Moving the map over $i$ into the contraction, we find

$$\mathcal{P}(f)(x, k) = \Sigma \left( b \mapsto \left( i \mapsto k(b) \frac{\partial f(x)(b)}{\partial x(i)} \right) \right) \tag{29}$$

$$= \Sigma(b \mapsto \mathcal{P}(x \mapsto f(x)(b))(x, k(b))) \tag{30}$$

$$= \Sigma(b \mapsto \mathcal{P}(\mathbf{C}(f)(b))(x, k(b))), \tag{31}$$

$$\mathcal{P}(\mathbf{C}(g))(x, k) = \Sigma(b \mapsto \mathcal{P}(g(b))(x, k(b))). \tag{32}$$

One may observe that the **C**-rule merely rewrites partial derivatives in terms of pullbacks and the index $i$ is eliminated. This rule does not perform any evaluation although the notation $g(b)$ makes is seem so. For example, the pullback of a map that double a vector element-wise can be derived without evaluation

$$\mathcal{P}(v \mapsto (i \mapsto 2v(i))))(v, k) = \mathcal{P}(\mathbf{C}(i \mapsto v \mapsto 2v(i)))(v, k) \tag{33}$$

$$= \Sigma(i \mapsto \mathcal{P}(v \mapsto 2v(i))(v, k(i))). \tag{34}$$

### 8.2 DIFFERENTIAING **B**

The last result in eq. (34) motivates us to find $\mathcal{P}(v \mapsto v(i))$. Following the definition of pullbacks in eq. (28), we have

$$\mathcal{P}(v \mapsto v(i)) = j \mapsto k(j) \frac{\partial v(i)}{\partial v(j)} = j \mapsto \delta(i, j, k(j)). \tag{35}$$

The result is a unit vector $\hat{e}_i k(i)$ if $i, j$ are integers. If $i, j$ are real numbers, the result is a "ket" $k(i) |i\rangle$ as used in quantum mechanics. There does not appear to be a common nomenclatures for the case where $i, j$ are paths. To obtain our **B**-rule, the derivation in eq. (35) can be generalized to $w \mapsto f(g)$, where $f$ is dependent on $w$

$$\mathcal{P}(w \mapsto f(g)) = j \mapsto k(b) \frac{\partial f(g)}{\partial w(j)} \tag{36}$$

$$= j \mapsto \Sigma \left( b \mapsto \frac{\partial f(b)}{\partial w(j)} \delta(b, g, k(b)) \right) \tag{37}$$

$$= j \mapsto \Sigma \left( b \mapsto \frac{\partial f(b)}{\partial w(j)} (i \mapsto \delta(i, g, k(i))(b)) \right) \tag{38}$$

$$= \mathcal{P}(w \mapsto f)(w, i \mapsto \delta(i, g, k(i))), \tag{39}$$

which is half of our **B** rule. The other half is the chain rule, which we will not prove because it is well-established.

What is less obvious is why the two parts can be simply added, so we will prove that now. Consider $\mathcal{P}(x \mapsto f(g(x)))$, where $f$ is dependent on $x$, we can write this as a different composition

$$\mathcal{P}(x \mapsto ((p, q) \mapsto p(q))((x \mapsto (f, g(x)))(x))). \tag{40}$$

$x \mapsto (f, g(x))$ is a function that returns a tuple. Its pullback can be derived from the eq. (28) as

$$\mathcal{P}(x \mapsto (f, g(x)))(x, k_1, k_2) = \mathcal{P}(x \mapsto f)(x, k_1) + \mathcal{P}(g)(x, k_2), \tag{41}$$

where the addition already arises in a similar fashion to partial derivatives. Likewise, $(p, q) \mapsto p(q)$ takes a tuple and applies the first object to the second. Its pullback is

$$\mathcal{P}((p, q) \mapsto p(q))(p, q, k) = (\mathcal{P}(p \mapsto p(q))(p, k), \mathcal{P}(q \mapsto p(q))(q, k)) \tag{42}$$
$$= (j \mapsto \delta(q, j, k(j)), \mathcal{P}(p)(q, k)). \tag{43}$$

Applying the chain rule to eq. (40) gives us the complete rule

$$\mathcal{P}(x \mapsto \mathbf{B}(f)(g)(x))(x, k) = \mathcal{P}(x \mapsto (f, g(x)))(x, \mathcal{P}((p, q) \mapsto p(q))(f, g(x), k)) \tag{44}$$
$$= \mathcal{P}(x \mapsto (f, g(x)))(x, (j \mapsto \delta(g(x), j, k(j)), \mathcal{P}(f)(g(x), k))) \tag{45}$$
$$= \mathcal{P}(x \mapsto f)(x, j \mapsto \delta(g(x), j, k(j))) + \mathcal{P}(g)(x, \mathcal{P}(f)(g(x), k)))). \tag{46}$$

# 9 SEQUENTIAL ITERATION AND FIXED POINTS

## 9.1 SEQUENTIAL ITERATION

To support general sequential iterations such as a `for` loop, we need a generalization of $\mathbf{B}$, which we denote as $\bigwedge$ and defined as composing a sequence of functions

$$\bigwedge f = f(N) \circ f(N - 1) \circ \ldots \circ f(1), \tag{47}$$

where $f$ is assumed to map an integer to a function. The integer $N$ is encoded in the domain of $f$ when $f$ is defined. A concrete example is summation, which can be implemented as

$$v \mapsto \bigwedge (i \in [1, N] \mapsto (x \mapsto x + v(i)))(0). \tag{48}$$

For notational brevity, we use $\bigwedge f$ and $\bigwedge(i \mapsto f(i))$ interchangeably.

Differentiating $\bigwedge$ means differentiating $\bigwedge f$ with respect to $f$, which is not differentiating $(\bigwedge f)(x)$ with respect to $x$. Denoting the intermediates as

$$y(i) = \left(\bigwedge_{t=1}^{i} f(t)\right)(y0), \quad l(i) = \left(\bigwedge_{m}^{i} k \mapsto \mathcal{P}(f(N - m + 1))(y(N - m), k)\right)(k(y0)), \tag{49}$$

the result is

$$\mathcal{P}(\bigwedge) = j \mapsto \lambda \mapsto \Sigma_{y0} \delta(\lambda, y(j - 1), l(N - j)). \tag{50}$$

The proof is simply applying the $\mathbf{C}$-rule before inducting on the fanout part of the $\mathbf{B}$-rule

$$\mathcal{P}\left(f \mapsto y0 \mapsto \left(\bigwedge_{i}^{N} f(i)\right)(y0)\right)(f, k) \tag{51}$$

$$= \Sigma_{y0} \mathcal{P}(f \mapsto y(N))(f, k(y0)) \quad \text{apply } \mathbf{C}\text{-rule} \tag{52}$$
$$= \Sigma_{y0} \mathcal{P}(f \mapsto f(N)(y(N - 1)))(f, k(y0)) \tag{53}$$
$$= \Sigma_{y0} \mathcal{P}(f \mapsto y(N - 1))(f, \mathcal{P}(f(N)))(y(N - 1), k(y0)) \tag{54}$$
$$\quad + \mathcal{P}(f \mapsto f(N))(f, \lambda \mapsto \delta(\lambda, y(N - 1), k(y0)) \quad \text{apply } \mathbf{B}\text{-rule} \tag{55}$$
$$= \Sigma_{y0} \mathcal{P}(f \mapsto f(N - 2))(f, \mathcal{P}(f(N - 1)))(y(N - 2), \mathcal{P}(f(N))(y(N - 1), k(y0))) \tag{56}$$
$$\quad + \mathcal{P}(f \mapsto f(N - 1))(f, \lambda \mapsto \delta(\lambda, y(N - 2), \mathcal{P}(f(N))(y(N - 1), k(y0)))) \tag{57}$$
$$\quad + \mathcal{P}(f \mapsto f(N))(f, \lambda \mapsto \delta(\lambda, y(N - 1), k(y0))) \tag{58}$$
$$= \Sigma_{y0} \Sigma_i \mathcal{P}(f \mapsto f(N - i))(f, \lambda \mapsto \delta(\lambda, y(N - i - 1), l(i))) \tag{59}$$
$$= \Sigma_{y0} \Sigma_i \mathcal{P}(f \mapsto f)(f, j \mapsto \delta(j, N - i, \lambda \mapsto \delta(\lambda, y(N - i - 1), l(i)))) \tag{60}$$
$$= \Sigma_{y0} (j \mapsto \Sigma_i \delta(N - j, i, \lambda \mapsto \delta(\lambda, y(N - i - 1), l(i)))) \tag{61}$$
$$= j \mapsto \lambda \mapsto \Sigma_{y0} \delta(\lambda, y(j - 1), l(N - j)). \tag{62}$$

Thus, the derivative depends on $y$ and $l$, which are the forward and backward intermediates in neural network or ODE settings. To differentiate a specific problem, one only needs to plug in a concrete $f$. For example, a `for` loop can be translated to $\bigwedge$ by explicitly passing the state as a variable, as shown in listing 6, which is a standard technique in functional programming for avoiding mutations. Immutable code generally comes with a performance overhead, but this is not a consequence of our conversion, but a general limitation of automatic differentiation.

Listing 6: Explicit state passing. The loop variable `i` is converted to be the first input of $f$ and the `state` are the second. The mutated state are returned as the output of $f$.

```
state = 0
for i in 1:10
    state = state + 1
end

final_state = seq(
(i::N{10}) ->
    state -> state + 1
)(0)
```

## 9.2 Fixed point

Our sequential iteration only works if the number of iterations is "stable", which means that it only depends on the sizes of our tensors and not the values within it. This is mostly acceptable in numerical applications with the main exception being the fixed point. Differentiating a fixed point is also known as implicit differentiation or sensitivity analysis. Fortunately, this special case is relatively straightforward to address.

Let us consider a system of equations $g(x) = 0$ and a procedure $\rho$ that maps $g$ to one of its roots, where the condition $g \mapsto g(\rho(g)) = g \mapsto 0$ is satisfied. Differentiating both sides yields

$$0 = \mathcal{P}(g \mapsto \rho(g))(g, \mathcal{P}g(\rho(g), k)) + \mathcal{P}(g \mapsto g)(g, i \mapsto \delta(i, \rho(g), k)) \tag{63}$$

$$= \mathcal{P}(\rho)(g, H(k)) + i \mapsto \delta(i, \rho(g), k), \quad H(k) = \mathcal{P}g(\rho(g), k) \tag{64}$$

$$\mathcal{P}(\rho) = (g, k) \mapsto i \mapsto \delta(i, \rho(g), -H^{-1}(k)), \tag{65}$$

where $H(k)$ is a linear map on $k$. The inverse of $H$ is a linear least square and can be written in terms of $\rho$ as

$$H^{-1}(k) = \rho(t \mapsto H(t) - k), \tag{66}$$

which does not require introducing new primitives and can be further differentiated.

There is a number of problems that can be treated as a fixed point for differentiation purpose. For example, minimization problems $\min_x R(x)$ can be treated as $\nabla R(x) = 0$. Constrained optimization problem can be treated by adding the constraints to $g$. Eigenvalue problems can may be treated as $g(\lambda, v) = Av - \lambda v$. Although a fixed point theory is too weak a formalism for most of these problems , it suffices for differentiation because we assume that the true solution has been found.

## 10 Relation to Source Transform

Let us illustrate the relation through a concrete example. Consider summing over a vector $\sum_{i=1}^{N} w(i)$, whose pullback can be derived as

$$\mathcal{P}(w \mapsto \sum_i w(i))(w, k) = \sum_i \mathcal{P}(w \mapsto w)(w, j \mapsto \delta(j, i, k)) = j \mapsto k.$$

The result is a vector whose elements are $k$.

If one were to do this the procedural way, the code would resemble listing 7.

Listing 7: Implementing summation as a for loop

```
sum = 0
```

```
for i in 1:N
    sum += w[i]
end
```

One can also unroll the ordered loop into a sequence of steps as in listing 8

Listing 8: Implementing summation as an unrolled loop.

```
w -> begin
    y1 = (t -> t + w(1))(0)
    y2 = (t -> t + w(2))(y1)
    ...
end
```

In either form, one can apply the standard prescription for source transformation as on page 125-127 in Griewank and Walther (2008).

The confluent functional counter part for the procedural sum is

$$\mathcal{P}(w \mapsto \bigwedge(i \mapsto t \mapsto t + w(i))(0))(w, k).$$

The differentiation is just repeated applications of **B**, **C**, and $\bigwedge$-rules. The result is

$$\sum_t q \mapsto \sum_{y0} \delta(t, y(q-1), l(N-q)) = q \mapsto \sum_{y0} l(N-q).$$

$l(N-q)$ are

$$l(N-q) = \bigwedge_{m=1}^{N-q} (k \mapsto \mathcal{P}(t \mapsto t + w(N-m+1))(y(N-m), k))(\delta(y0, 0, k))$$

This can be simplified into $\bigwedge_{m=1}^{N-q}(k \mapsto k)(\delta(y0, 0, k)) = \delta(y0, 0, k)$. Therefore, the final result is again $q \mapsto k$. However, if one now unroll $y$ and $l$, the result is shown in listing 9, which is almost the same as the result of regular source transform.

Listing 9: Unrolled derivative of summation.

```
w -> sum(y0 -> begin
    y1 = (t -> t + w(1))(y0)
    y2 = (t -> t + w(2))(y1)
    y3 = (t -> t + w(3))(y2)
    l0 = delta(y0, 0, k)
    l1 = P(t -> t + w(3))(y2, l0)
    l2 = P(t -> t + w(2))(y1, l1)
    [l2, l1, l0]
    end)
end
```

The `delta` can propagate out and annihilate with the `y0` sum because pullbacks are linear. Then we will have completed a source transform.

We can immediately generalize the problem to differentiating a model $x \mapsto \ldots$ with respect to model parameters $w$

$$\mathcal{P}(w \mapsto x \mapsto \bigwedge(i \mapsto t \mapsto t + w(i) * x(i))(0)),$$

which is sometimes the point where typical source transform systems start to struggle. In our framework, the derivation become an extra application of the C rule.

## 11 COMPLEX PRIMITIVES

We now explain our treatment of complex numbers, which leads to the complex conjugates in primitive pullbacks. The main difficulty in dealing with complex numbers is that the

standard complex analysis does not prescribe a useful gradient for optimization. For example, minimizing $z \mapsto |z|^2$ is evidently equivalent to minimizing $(a, b) \mapsto a^2 + b^2$, but a Cauchy-Riemann argument shows that $|z|^2$ is nowhere analytic, so the pullback makes no sense. For a real and scalar valued function, this problem is partly resolved through the Wirtinger derivative

$$\partial f(z)/\partial z = \partial f(z)/\partial a + i\partial f(z)/\partial b, \quad z = a + ib, \tag{67}$$

which can be used for, e.g., gradient descent.

This formalism is insufficient for symbolic automation because it does not handle the case where $f(z)$ is complex and requires splitting $z$ into its real and imaginary parts. Differentiating a complex function $f(z)$ may seem unnecessary when the objective function to optimize is always a real scalar. However, we differentiate $f(z)$ by differentiating its constituents, which are complex-valued functions. Moreover, representing a complex gradient in terms of the real and imaginary parts of $z$ is not acceptable for symbolic purposes, and it is preferable to avoid splitting a complex variable to begin with (rather than trying to reassemble them from the real and imaginary parts in the end).

These problems can be resolved by extending the definition of pullback to complex numbers. We start by proposing the operators $\mathcal{V}$ and $\mathcal{W}$

$$\mathcal{V}(z) = [\mathrm{Re}(z_1) \quad \mathrm{Im}(z_1) \quad \ldots \quad \mathrm{Re}(z_n) \quad \mathrm{Im}(z_n)]^T, \tag{68}$$

$$\mathcal{W}(f) = v \mapsto \mathcal{V}(f(\mathcal{V}^{-1}(v))). \tag{69}$$

$\mathcal{V}$ and $\mathcal{V}^{-1}$ establish an isomorphism between $\mathbb{C}^N$ and $\mathbb{R}^{2N}$ so that we can convert a complex problem to a real one that is equivalent. Analogously, $\mathcal{W}$ converts between $\mathbb{C}^N \to \mathbb{C}^M$ and $\mathbb{R}^{2N} \to \mathbb{R}^{2M}$. One can check that the following identities hold

$$\forall f \in \mathbb{C}^N \to \mathbb{C}^M, \quad \mathcal{V}(f(z)) = (\mathcal{W}(f))(\mathcal{V}(z)), \tag{70}$$

$$\forall f \in \mathbb{C}^N \to \mathbb{R}, \quad f(z) = \mathcal{V}(1)^T \cdot (\mathcal{W}(f))(\mathcal{V}(z)). \tag{71}$$

To minimize a scalar-valued function $f(z)$ over $z$, we can equivalently minimize the real function $u \mapsto \mathcal{V}(1)^T \cdot (\mathcal{W}(f))(u)$ and convert $u$ to the corresponding complex number with $z = \mathcal{V}^{-1}(u)$. The gradient of the real function can be written as $(\mathcal{J}(\mathcal{W}(f)))(u)^T \cdot \mathcal{V}(1)$. Transforming this vector back into the complex space gives the complex gradient $\mathcal{V}^{-1}\left((\mathcal{J}(\mathcal{W}(f)))(u)^T \cdot \mathcal{V}(1)\right)$. Therefore, we write the Wirtinger gradient as

$$\nabla f(z) = \mathcal{V}^{-1}(\mathcal{J}(\mathcal{W}(f))(\mathcal{V}(z))^T \cdot (\mathcal{V}(1))). \tag{72}$$

To be able to find the gradient through the pullback as $\nabla f(z) = \mathcal{P}(f)(z, 1)$, we suggest to define the complex pullbacks as

$$\mathcal{P}(f) = (z, k) \mapsto \mathcal{V}^{-1}(\mathcal{J}(\mathcal{W}(f))(\mathcal{V}(z))^T \cdot \mathcal{V}(k)). \tag{73}$$

Since the pullback remains a vector Jacobian product just like eq. (28), the **B** and **C** rules are not affected by the change. Therefore, the only modification to the theory is to derive the pullbacks of the univariate primitives using eq. (73) instead of eq. (28). As an example, writing $z = x + iy$ and $k = a + ib$, the pullback of the complex conjugate can be derived as

$$\mathcal{W}(z \mapsto z^*) = (x, y) \mapsto (x, -y), \tag{74}$$

$$\mathcal{P}(z \mapsto z^*) = (z, k) \mapsto \mathcal{V}^{-1}\left(\begin{bmatrix} 1 & 0 \\ 0 & -1 \end{bmatrix} \cdot \begin{bmatrix} a \\ b \end{bmatrix}\right) = (z, k) \mapsto k^*. \tag{75}$$