# OpenReview forum: "Differentiating without Partial Evaluation"
_ICLR.cc/2026/Conference — Submitted to ICLR 2026_

### Official Review · Reviewer_reKi · 2025-10-25

**Soundness:** 4
**Presentation:** 3
**Contribution:** 2
**Rating:** 2
**Confidence:** 5

**Summary:**

This paper proposes applying reverse mode automatic differentiation (AD) to functional programs expressed as combinators, rather than traditional methods applied to imperative programs or functional programs expressed in variants of the lambda calculus. The key contribution is deriving formulas to differentiate the B and C cominators, B denoting function composition and C denoting swapping the first two arguments of a curried function. With this, AD becomes syntactic and trivial: basis functions f are replaced with their reverse mode transformations f', B is replaced with its reverse mode transformation B', and C is replaced with its reverse mode transformation C'.

I haven't slogged through the math. But I will give the authors the benefit of the doubt. If there are errors, they must be minor and can be easily fixed as in principle it is possible to do what is claimed in the submission. But it is not necessary to do so, given my comments below.

**Strengths:**

The general idea of applying AD to combinators is sound and novel. The reverse mode transformation of B is well known and trivial. It is the chain rule. It has been presented in many places, among others in Pearlmutter & Siskind (TOPLAS 2008). The reverse mode transformation of C is novel but straightforward.

It would be great to see the general approach fleshed out to all combinators, a Turing complete set of combinators, or at least a more powerful set of combinators. This is not done in this paper. I encourage the authors to do so.

It would be great to see this turned into a practical and efficient AD system that is competitive with the likes of PyTorch and JAX, one that could generate efficient gradients of arbitrary code written in an inhabitable functional programming language that ran competitively on GPUs.

I encourage the authors to continue this line of work to flesh out the above.

**Weaknesses:**

The general claim that all other approaches to (reverse mode) AD require tracing/partial evaluation is false. Forward mode AD using dual numbers does not required tracing. Many classical AD systems, like Adifor and Tapenade do source-to-source transformation for forward and/or reverse mode AD without tracing or partial evaluation. This has been done at least since the JAKE system, Speelpenning (1980). Even for functional programming, Pearlmutter & Siskind (TOPLAS 2008) did this for the untyped lambda calculus. Many follow-on authors elaborated on this. These methods handle Turing complete languages. Since, one can formulate B and C as trivial lambda-calculus expressions, the results in this submission trivially follow from prior work.

The B and C combinators are not Turing complete. They are not even very powerful. You cannot write map and reduce in them. You cannot even swap other than the first two arguments of a function.

The key limitation of this work is that neither B nor C involve fanout. They are both linear operators. Reverse-mode AD is trivial for linear operators. Reverse-mode AD becomes difficult when there is fanout because you need to handle accumulation. Fanout is needed to handle combinators such as S and Y.

The appendix gives a method for transforming what the authors call stable iterate-to-fixed-point operators, i.e. ones that have a number of iterations are not dependent on floating point values There has been work on transforming general iterate-to-fixed-point operators.

**Questions:**

None

---

> ### Author Response · Authors · 2025-11-21
>
> We appreciates the insightful feedback from the reviewer. In particular, the fanout is a very important remark. We also like to thank the reviewer for recognizing the interesting aspects of this approach, and fleshing out the work to other combinator would be interesting.
>
> "AD require tracing/partial evaluation"
>
> We did not make a general claim. We will rephrase this point further in the next revision. Please see the general comment for clarification.
>
> "The reverse mode transformation of B is well known and trivial. It is the chain rule."
>
> The B rule has two components. The first part is the trivial chain rule but the second has fanout in it. Reviewer zHBj pointed out that the B rule has appeared in previous literature Ehrhard & Regnier (2003).
>
> "The key limitation of this work is that neither B nor C involve fanout."
>
> This is a very insightful remark. We use B with captures, which appears to be fundamentally built from S.
> For example, the binary product rule can be derived from the unary product rule as (keeping both two operants $x$ for simplicity).
>
>   P(x→x∗x)
>
> =P(x→(v→x∗v)(x))
>
> =(x,k) →P(x→x)(x,P(v→x∗v)(x,k)) + P(x→(v→x∗v))(x,i→δ(i,x,k)) (chain rule + fanout)
>
> =(x,k) →P(v→x∗v)(x,k) + ∑ᵥP(x→x∗v)(x,δ(v,x,k))
>
> =(x,k) →x k+∑ᵥv δ(v,x,k)
>
> =(x,k) →2xk.
>
> "The B and C combinators are not Turing complete. They are not even very powerful."
>
> This is true and worth pointing out. We will include a discussion in the next revision.
>
> "You cannot write map and reduce in them."
>
> Maps are implemented in our DSL as an "impatient" evaluation strategy instead of a function.
> This means, functions whose input types are indices have all their values stored in a tensor, thus representing a tensor.
> Therefore, `y = map(f, xs)` is implemented as `y = (i) -> f(xs(i))`, which can be inlined (`y(1)` can be compiled to `f(xs(1))`) to save memory. The C rule allows for the differentiation of map although it does not encode it. For example, consider the identity map
> from a vector to itself $x \mapsto (i) \mapsto x(i)$. We can differentiate it with C rule combined with the fanout part of B rule.
> $$
> \mathcal{P}(x \mapsto (i \mapsto x(i)))(x, k) = \sum_i \mathcal{P}(x \mapsto x(i))(x, k(i))
> = \sum_i (j) \mapsto \delta(i, j,  k(i)) =j \mapsto k(j))  = k
> $$
>
> In the appendix, we derived a slightly generalized B rule for  $\mathcal{P}(f \mapsto \circ_{i=1}^N f(i))$, which is the pullback of the map from a vector of functions to their composition. The result is derived by an inductive application of the B and C rules, and the fanout part of B recovers the accumulation of reverse-mode AD. Please see our response to reviewer qSzs for an exposition of how this theory can differentiate mutations and how it implies source transform AD. At this point, given the sharp insight of the reviewer, it shouldn't be hard to see that `foldr` can be implemented, although unordered evaluation of reduce is less straightforward. However, the only commutative use of reduce we have encountered in physical sciences so far is sum and prod, which are built in to the DSL for symbolic processing reasons.
>
> "It would be great to see this turned into a practical and efficient AD system"
>
> We thank the reviewer for the kind encouragement. The DSL is intended as a frontend, which is to be compiled to numerical primitives at various levels for performance.
> Please see the general comment on toolchain for more details.

---

### Official Review · Reviewer_qSzs · 2025-10-29

**Soundness:** 2
**Presentation:** 3
**Contribution:** 2
**Rating:** 6
**Confidence:** 5

**Summary:**

The paper “Differentiating without Partial Evaluation” introduces an approach to symbolic differentiation that avoids the traditional reliance on partial evaluation, primarily through an alternative use of combinatory logic. The authors extend standard differentiation beyond the chain rule via a second rule derived from B/C combinators allowing gradients to be derived without any partial evaluation of the program.  The authors argue this avoids expression swell and keeps gradients symbolic until the very end, contrasting with Auto Diff systems that first evaluate or trace to a graph. This perspective on framing differentiation around a complete combinator basis and giving explicit pullback rules for B and C is an original synthesis in ML/AD literature.
The paper provides a good survey of existing literature on this topic.
The report is largely complete in scope for a theoretical proposal. It provides sufficient background, explains the motivation and theory behind the second differentiation rule, shows its application through concrete examples (MLP gradients, Hartree-Fock energy, conjugate gradient optimization), and discusses both theoretical and practical limitations. However, its primary implementation is a proof-of-concept within a domain-specific language (Julia), and lacks comprehensive benchmarks across mainstream tools or large-scale problems.  This omission of demonstration against large scale problems reduces confidence in the direct and broad use and benefit of this approach.
In this paper, all symbolic differentiation is done at compile time, using pullback rules for the B and C combinators. This means that the differentiation process operates on program structure, not runtime values and relies on statically typed, dimensionally fixed programs, where all tensor shapes and types are known ahead of time.  This can be a limitation when dealing with NN and physical simulations.  The authors acknowledge this limitation.

**Strengths:**

Paper is well motivated and the developments are sound and well structured.
The paper presents a  conceptual advancement completing the chain rule with a second rule to avoid partial evaluation; symbolic-first gradients.
The benefits of this approach include Interpretability & symmetry.  Gradients remain symbolic, enabling physics-aware simplification.
Illustrative examples across multiple domains such as NN, quantum chemistry tensors, and numerical linear algebra provide confidence.
Proofs provided in the appendix are adequate.

**Weaknesses:**

No performance evaluation: no runtime/memory or large-scale benchmarks vs. JAX/Enzyme/SymPy; toolchain impact remains unquantified (acknowledged by authors)
Practical limits: needs stable dimensions, lacks mutation support, and fixed-point/sequential iteration support is only sketched (appendix); these are crucial for many scientific codes.
Integration hurdles: static typing and compile-time differentiation requirement may clash with dynamic Python/Julia ecosystems; engineering path is non-trivial
Accessibility: combinatory-logic framing may be unusual for many ML/AD practitioners, increasing the learning curve.
The limitations related to 1. Dimensional Stability, 2. Fixed point iteration gaps, 3, lack of mutation support appear to be  a direct consequence of the framework that relies on pure symbolic diff (no tracing or evaluation) , while this provides certain benefits, in my opinion this also imposes restriction to purely functional, statically shaped, non-iterative programs.  In that I agree with  the authors that this work is paper is more a proof of concept of a symbolic calculus and not a production-ready AD method.

**Questions:**

Claim: “complete the chain rule” with a second rule based on B and C combinators.
1. Is this new rule provably complete for all differentiable compositions expressible in combinatory logic?
2. Can every standard AD operation be represented equivalently under your B/C pullback formulation?

How does the symbolic calculus relate formally to reverse-mode AD or the dual number formulation?
Can the framework be seen as a category-theoretic dual of reverse-mode AD (e.g., functorial composition)?


Given differentiation without partial evaluation, how does one ensure equivalence to the derivative of the evaluated program rather than the unevaluated syntax tree?

A formal criterion for when symbolic contraction (via delta identities) terminates in closed form would be useful.

Would integrating this calculus into a JIT or AOT compiler break referential transparency, and if so, how could this be mitigated?

**Details Of Ethics Concerns:**

no ethical concerns

---

> ### Author Response · Authors · 2025-11-22
>
> We thank the review for the thoughtful comments and questions. We will take the
> opportunity to expand the relationship to source transform AD.
>
> "performance and toolchain"
>
> Please refer to the general comment.
>
> "mutation and iteration"
>
> We will expand on this below.
>
> "Is this new rule provably complete"
>
> We do not have a proof, although source transform AD is implied as shown below.
>
> "Can every standard AD operation be represented"
>
> There does not seem to be an exhaustive list, so we will demonstrate how it is done for a mutation.
>
> First, we bring the mutation into a confluent functional form. For example, `x[i]
> += 1` is confluent to $x = (j) \mapsto x(j) + \delta(i, j, 1)$, which maps $x$
> to an updated new vector. The functional form can be differentiated first by applying
> the C rule
> $$
> \mathcal{P}(x \mapsto j \mapsto x(j) + \delta(i, j, 1))(x, k)
> =\sum_j \mathcal{P}(x \mapsto x(j) + \delta(i, j, 1))(x, k(j))
> $$
> Then we apply the B rule to get
> $$
> \sum_j i \mapsto \delta(i, j, k(j)) =i \mapsto k(i) = k.
> $$
>
> Therefore, we have concluded that incrementing an array element by a constant
> merely adds an identity map in the backpropagation.
>
> "How does the symbolic calculus relate formally to reverse-mode AD"
>
> The differentiation rule for $\bigwedge$, which is just repeated B (or perhaps
> S as informed by reviewer reKi's comment) is basically reverse-mode source transform as
> described on page 125-127 in [this reference
> book](https://epubs.siam.org/doi/book/10.1137/1.9780898717761), except that we
> encode it as a formula.
> The benefit of a formula over a prescription is that it is easily composed, as we will show towards the end.
> Functionally, $\bigwedge$ composes a vector of functions with
> captures. The rule is reasonably concise and proved in the appendix
> $$
> \mathcal{P}\left(\bigwedge\right) = (f, k) \mapsto
> j \mapsto \lambda \mapsto \sum_{y0}
> \delta(\lambda, y(j - 1),
> l(N - j)
> ),
> $$
> where $y$ are the regular variables and $l$ are analogous to the dual
> $$
> y(n) = \left(\bigwedge_i^{n} f(t)\right)(y0), \quad
> l(i) =  \left(\bigwedge_m^{i} (
> k \mapsto \mathcal{P}(f(N - m + 1))(y(N - m), k))\right)(k(y0)).
> $$
>
> Let us illustrate the relation through a concrete example. Consider summing
> over a vector $\sum_{i = 1}^N w(i)$, whose pullback can be derived as
> $$
> \begin{aligned}
> \mathcal{P}(w \mapsto \sum_i w(i))(w, k)
> =\sum_i \mathcal{P}(w \mapsto w)(w, j \mapsto \delta(j, i, k))
> = j \mapsto k.
> \end{aligned}
> $$
> The result is a vector whose elements are $k$.
>
> Let us now do this the procedural way. The confluent functional counterpart is
> $$
> \begin{aligned}
> \mathcal{P}(w \mapsto \bigwedge(i \mapsto t \mapsto t + w(i))(0))(w, k).
> \end{aligned}
> $$
> One can unroll the ordered loop into procedural code as
> ```
> w -> begin
>     y1 = (t -> t + w(1))(0)
>     y2 = (t -> t + w(2))(y1)
>     ...
> end
> ```
>
> The differentiation is just repeated applications of B, C, and $\bigwedge$
> rules. Due to character limit, we will just present the result
> $$
> \sum_t q \mapsto \sum_{y0} \delta(t, y(q - 1), l(N - q))
> = q \mapsto \sum_{y0} l(N - q),
> $$
> where
> $$
> l(N - q) = \bigwedge_{m = 1}^{N-q}(k \mapsto \mathcal{P}
> (t \mapsto t + w(N - m + 1))(y(N-m), k))(\delta(y0, 0, k))
> $$
> This can be simplified into
> $\bigwedge_{m = 1}^{N-q}(k \mapsto k)(\delta(y0, 0, k)) = \delta(y0, 0, k).$
> Therefore, the final result is again $q \mapsto k$. However, if one now unroll
> $y$ and $l$, the result is almost the same as source transform
> ```
> w -> sum(y0 -> begin
>     y1 = (t -> t + w(1))(y0)
>     y2 = (t -> t + w(2))(y1)
>     y3 = (t -> t + w(3))(y2)
>     l0 = delta(y0, 0, k)
>     l1 = P(t -> t + w(3))(y2, l0)
>     l2 = P(t -> t + w(2))(y1, l1)
>     [l2, l1, l0]
>     end)
> end
> ```
> The `delta` can propagate out and annihilate with the `y0` sum because
> pullbacks are linear. Then we will have completed a source transform.
>
> The benefit of having rule is that we can immediately generalize the problem to differentiating a model
> $x \mapsto \ldots$ with respect to model parameters $w$
> $$
> \begin{aligned}
> \mathcal{P}(w \mapsto x \mapsto \bigwedge(i \mapsto t \mapsto t + w(i) * x(i))(0)),
> \end{aligned}
> $$
> which is sometimes the point where one resorts to tracing. In our framework,
> the derivation becomes an extra application of the C rule.
>
> "ensure equivalence"
>
> We are unsure about what this question means. Could you elaborate?
>
> "formal criterion"
>
> We agree that it would be useful, but we do not have a formal criterion yet.
>
> "JIT or AOT"
>
> When defining a new function such as `y -> i -> y(i) * x(i)` in our DSL, we
> receive a pure function `f = x -> y -> i -> x(i)` and the value of the free
> variable `x` captured from the defining scope.
> The pure function `f` will be jitted and the users will manage
> the context `x` as they see fit.

---

### Official Review · Reviewer_zHBj · 2025-11-01

**Soundness:** 3
**Presentation:** 3
**Contribution:** 2
**Rating:** 4
**Confidence:** 4

**Summary:**

The paper proposed a framework for programmatic transformation of mathematical codes to their derivatives. The main difference from other AD systems is that it adopts the tacit programming style (also referred to as point-free style), where the evaluation of variables (which can represent functions) is deferred to the very end. In contrast, in most AD, one must substitute the variables/symbols representing functions with concrete instances before tracing. This is possible as the proposed framework represents programs in a DSL based on combinatory logic. In combinatory logic, there are only combinatory terms (function primitives) and combinators (higher-order functions), and it is proven to be Turing-complete. Thus, a program can be written as a transformation of primitive functions under the combinators. Usual AD only defines the pullback on function primitives, but in this framework, the pullback for the combinators, which enable one to programtically differentiate combinatory logic. It is well known that the B and C combinator forms a complete basis, so one only needs to define the pullback (VJP) for B and C, which is provided in the paper. The paper also illustrates how a tensor program can be represented using combinatory logic. Finally, the paper demonstrates how the proposed system can be used to: (a) compute the gradient of MLP with respect to network parameters; (b) compute the HF gradient with symmetry; (c) derive conjugate gradient using the symbolic gradient output from the proposed system.

**Strengths:**

The paper provides an interesting alternative to the common AD system, where one can add in symbolic simplification rule easily (which is demonstrated in the HF gradient example).
The paper is well written and easy to follow, and is relatively self-contained.

**Weaknesses:**

My main concern is that the author failed to convey how the proposed B/C‑based differentiation differs, in capability and guarantees, from prior functional/program‑transform approaches to AD. All 3 examples demonstrated in the paper can be done with existing frameworks:
- MLP gradient: this is quite standard and can be computed in JAX/PyTorch easily
- Symmetry in HF gradient: one should note that this is a highly specialized application. I'd argue that using something like jax.custom_vjp suffices. Even within the proposed framework, one would first need to code the symmetry rule into the framework, so I can hardly see the benefit
- symbolic derivation of CG: this is neat, but mainstream CAS like Maple, Mathematica, Sympy, etc can also do this easily.

Evidence on a curated suite of "hard" patterns that cannot be done within the existing AD/CAS framework would help.

The paper claims that it avoids expression swell, which symbolic AD is susceptible to. But the paper does not provide any concrete evidence against an existing symbolic AD system, nor does it provide a theoretical justification for the claim.

Also, the paper specifically states in the introduction section that the paper only makes qualitative claims. This actually weakens the potential impact of the paper, since without numerical evidence, the benefits of the proposed framework are at best speculative. There are no quantitative evaluations: no expression‑size statistics, compile times, runtime/peak‑memory comparisons vs. tracing AD (PyTorch/JAX), source transformation (Enzyme), or symbolic systems (SymPy/Mathematica). Even small benchmarks (MLP with/without accumulation; HF with symmetry folding; end‑to‑end cost of simplification + engine execution) would significantly strengthen the claims.

Finally, I would like to point out that the idea is not entirely new. The core contribution of the paper appears to be a point-free style symbolic differentiation technique that uses the pullback rule of the B and C combinator. I would like to point out the following prior works that partially encompass the main idea of the paper:
- point-free / higher-order AD: there are many such systems in the functional programming community. For example, Haskell's AD, and [The Differentiable Curry: https://openreview.net/pdf?id=ryxuz9SzDB].
- Using combinatory logics to formulate AD: see [Elsman et al. (2022), Combinatory Adjoints and Differentiation]
- B-rule: [Ehrhard & Regnier (2003), differential λ-calculus].
- C combinator swaps the order of currying, which is also known as the "flip" combinator. The C-rule in this paper is a direct consequence of the fact that the C combinator commutes with derivative operators, which is established much more rigorously in prior works, e.g., [Blute, Cockett & Seely, Cartesian Differential Categories] and this PhD thesis [https://cspages.ucalgary.ca/~robin/Theses/GallagherPhD.pdf].
- representing tensor as a function from index to value: this is a common practice, which is already adopted in TVM/TACO/Mathematica, etc.

**Questions:**

- It seems that it is also possible to define the pushforward for the B and C combinator. Have you thought about it?

---

> ### Author Response · Authors · 2025-11-21
>
> We appreciate the reviewer for providing the thoughtful feedback and relevant
> references, both of which helped us navigating the problem.
>
> "Difference from prior approaches"
>
> The advantage is to bring the three frameworks mentioned by the reviewer
> (algorithmic, by hand, and symbolic) into one system, so that scientists do not
> have to accept their trade-offs as much. The first use case is in producing a symbolic
> gradient within an AD system so that adopting AD is on the path to a
> hand-optimized gradient. The second reason is to potentially let
> the compiler optimize over a large number of high-level primitives given the
> symbolic gradient without having to maintain a library of vjp rules.
> The Hartree-Fock example illustrates this point. The difficulty is not in extending
> an AD system to get the gradient right once we know how, but in finding out what primitives to
> use and what the vjp rule should be, which requires observing the symbolic gradient to know and automate.
> For a more detailed discussion, please refer to the evaluation section in our general comment
>
>
> The Hartree-Fock example is in fact representative of an important class of
> problems that are hard for existing AD/CAS systems. A rather large number of
> problems in physical sciences including quantum chemistry, condense matter, and
> material simulation involve large tensor operations over complex numbers with
> some combination of symmetries. The point of specifying the symmetries instead of the vjp rules is that the symmetries of the tensors are few and known, but each combination of different symmetries in a tensor operation leads
> to a different vjp rule, and there is not a finite number of different tensor
> operations. The standard practice for solving these problems
> is still differentiation by hand due to obscure performance in AD case studies.
> For example, a recent [effort](https://docs.dftk.org/stable/examples/forwarddiff/) to leverage AD for DFT ended up using forward mode for $\mathbb{R}^N \to \mathbb{R}$ problems even with $N = 12$ because reverse mode turned out slower.
>
> "expression swell"
>
> This is a fair point. We should revise the claim to restrict to common cases of
> expression well within physical sciences, where problems originate from
> discretizing a PDE or an ODE. For differentiation purposes, the mathematics
> here can be reduced to a fixed point map or time integration, which can be
> viewed as an higher order function that take one or more large tensor operations as input.
> Expression swell happens only through symbolic evaluation. In scalar operations,
> this can happen due to symbolic evaluation of functions. For example $(a + b)^n
> \to a^n + b^n + \ldots$, which is easily solved with the chain rule.
>
> In tensor operations, expressions swell when replacing a symbol with an array
> of symbols $x \to [x_1, x_2, \ldots, x_N]$ for the purpose of reducing the
> problem to univariate differentiation. The C rule and the second part of B rule
> solves this problem. For example, consider the identity map of a vector $x
> \mapsto (i \mapsto x(i))$, the C rules and the second part of B rule produces
> $$
> \mathcal{P}(x \mapsto (i \mapsto x(i)))(x, k) = \sum_i \mathcal{P}(x \mapsto x(i))(x, k(i))
> = \sum_i (j) \mapsto \delta(i, j,  k(i)) =j \mapsto k(j))  = k
> $$
> Therefore, we have derived the pullback of the tensor operation without writing
> out a Jacobian matrix or making a new vjp rule.
>
> The solution to fixed point and time integration are derived from the B and C
> rules in the appendix. The result for fixed point is equivalent to existing
> techniques for differentiating [bi-level
> optimization](https://arxiv.org/abs/1607.05447). The result for time
> integration loosely recovers tracing-based AD when unrolled, but we are unsure if the
> abstract formula has been known.
>
> "numerical evidence"
>
> Please refer to the general comment for why we do not evaluate. Our DSL is a
> frontend that provides symbolic derivatives for tensor operations as well as
> MLP-like applications. The primary metric for evaluating a frontend is user
> effort, which is also arguably the deciding factor in choosing amongst
> existing AD systems because they provide similar performance if carefully tuned.
> Because human effort is difficult to quantify, we believe that qualitative
> arguments based on representative examples are the more informative evaluation.
> Moreover, the numerical performance depends entirely on the backend, which
> can in principle be Enzyme or PyTorch if desired, so a performance evaluation against them
> would not reveal much.
>
> "prior works"
>
> We appreciate the reviewer for these informative references. The third point is
> particularly relevant. We will include them in a revision.
>
> "pushforward for the B and C combinator"
>
> We have not derived it, but it should not be difficult.

---

### Official Review · Reviewer_AXQM · 2025-11-04

**Soundness:** 1
**Presentation:** 2
**Contribution:** 1
**Rating:** 2
**Confidence:** 5

**Summary:**

In the present paper, the authors introduce an additional differentiation rule to complement the chain-rule used in automatic differentiation systems. Theoretically deriving its results, the application of the rule to skip the partial evaluation is demonstrated on a set of code examples.

**Strengths:**

The strength of this paper lies in its writeup, and the intellectual clarity with which its ideas are presented. The derivations are done with great care, and its chosen code examples help to present its ideas and aid in understanding the application of the differentiation rule.

**Weaknesses:**

The weaknesses of the paper are plentiful, distilling into high-level topics before diving into them individually:
- Misguided, or ill-informed key assumptions about automatic differentiation systems to motivate the current work
- The _complete_ lack of evaluation of the current work

#### Misguided Key Assumptions
- line 16-17 "gradients irreversibly become blackboxes", there exist a wide range of automatic differentiation systems in practice. The key difference being abstraction level chosen by the automatic differentiation system. As such this core claim of the motivation does not stand up to scrutiny. Especially for source transformation AD systems such as Tapenade [1], the gradients are emitted in the source language for the user and can be inspected. But even the big modern AD systems JAX, and PyTorch permit the emission of the produced gradients [2], which can then be inspected and are hence neither blackboxes, nor riddled with "obscure performance ceilings".
- The leveraging of symmetries, as envisioned by the authors, is something that is not impossible for modern automatic differentiation systems to perform. It is called custom rules. The mentioned Enzyme provides ample infrastructure for that, which can then leverage physical symmetries [3].
- Line 48, the PyTorch citation is pointed at Baydin et al. This is wrong. The citation should read Paszke et al [4].

#### Lack of Evaluation
- The presented produced? code is never executed and as such it is not possible to quantify the performance benefits the additional differentiation rule could provide. Especially the Hartree Fork system would naturally lend itself to such evaluation where one could for example evaluate
    - Existing AD systems (Enzyme, PyTorch/JAX, other Julia AD systems such as Zygote for example)
    - The prototypical implementation of yours
    - Prototype of yours leveraging the available symmetries
- The presented code examples are not in Julia, the language in which the prototype implementation of this work is implemented in (line 244f), as such it is not apparent to the reviewer whether the proof-of-concept is leveraged in this work. Even if the prototype is not able to perform the differentiation of the presented illustrative examples, I would at a minimum expect an evaluation on AD micro benchmarks. See e.g. the tests of Enzyme [5] for a large number of micro-micro examples one could leverage here.

References:
1. Laurent Hascoet and Valérie Pascual. 2013. The Tapenade automatic differentiation tool: Principles, model, and specification. ACM Trans. Math. Softw. 39, 3, Article 20 (April 2013), 43 pages. https://doi.org/10.1145/2450153.2450158
2. https://docs.pytorch.org/tutorials/intermediate/inductor_debug_cpu.html
3. https://enzyme.mit.edu/julia/stable/#Defining-rules
4. Paszke, A., Gross, S., Massa, F., Lerer, A., Bradbury, J., Chanan, G., Killeen, T., Lin, Z., Gimelshein, N., Antiga, L., Desmaison, A., Köpf, A., Yang, E., DeVito, Z., Raison, M., Tejani, A., Chilamkurthy, S., Steiner, B., Fang, L., Bai, J., & Chintala, S. (2019). PyTorch: An Imperative Style, High-Performance Deep Learning Library. ArXiv, abs/1912.01703.
5. https://github.com/EnzymeAD/Enzyme/tree/main/enzyme/test/Enzyme/ReverseMode

**Questions:**

- I am a little confused by one of the key claims made by the abstract with relation to interpretable surrogates (line 12-14). While the ability to inspect the gradient flow is highly conducive to interpretable surrogates it is unclear to the reviewer how the present system provides any advantage here. As mentioned previously, Tapenade is able to emit the source code of the unevaluated gradient expression for inspection, and the reviewer would contend that the need for performance of the gradient expression supersedes the need for unevaluated gradient expressions (which can yet be inspected in all of the mentioned AD systems).
- It is entirely unclear to the reviewer whether the prototype implementation of the work is ever used throughout the paper? Would the authors be able to provide evidence that their prototype is actually functional?

**Details Of Ethics Concerns:**

/

---

> ### Author Response · Authors · 2025-11-21
>
> We thank the reviewer for the criticism.
>
> "gradients irreversibly become blackboxes"
>
> Our claim is predicated on a specific subtype of problems $\mathcal{P}(w \mapsto f(w)(g))$, which is usually solved with tracing and known to cause complications in source transformation systems as pointed out by reviewer zHBj in this [reference](https://openreview.net/pdf?id=ryxuz9SzDB). The authors are aware of source transformation systems and the fact that they emit source code, as we discussed in our introduction. We also did not claim that AD systems are "riddled" with the performance ceilings.
>
> "Custom rules"
>
> We are aware of custom vjp rules. The difficulty is not in putting it into an AD system, but in identifying where it is needed and deriving them, which is the point of the example.
>
> "PyTorch citation is pointed at Baydin et al. This is wrong. The citation should read Paszke et al [4]."
>
> Thanks for point it out. We will fix it.
>
> "Lack of Evaluation" and "micro benchmarks"
>
> Please see the general comment on why no performance evaluation is performed. Our DSL is a "frontend" that obtains the gradient in a symbolic form largely for its own sake. We leave the backend to existing primitives and the choice is left to (future) compiler optimization. We could try to obtain a fast gradient either by implementing the expression by hand or compiler optimization, but
> we could also lower the model to llvm and leave the rest to Enzyme. Therefore, performance comparison makes very little sense.
>
> "Code examples"
>
> The software package and all examples are submitted with the paper as a supplement. The code is able to produce the presented output, which is not numerically evaluated.
>
> "it is unclear to the reviewer how the present system provides any advantage here"
>
> The present system provides a symbolic gradient for tensor operations. For example, we write a matrix vector product as
> `x -> (i) -> sum(j->A(i, j) * x(j))`, which is differentiated into `(x, k) -> (i) -> sum(j->A(j, i) * k(j))` without rules for matrix calculus.
> This is different from a double for loop implementation, which is not useful for further symbolic reasoning and the compiler cannot identify it with a BLAS call.
>
> The system also can handle captures and give an analytic answer to $\mathcal{P}(w \mapsto f(w)(g))$. This is necessary for differentiation with respect to model parameters, which includes weights of neural network and controls of differential equations. Our understanding is that early source transformation systems do not address this problem, which contributed to the rise of tracing. We suspect that newer systems can do this, but the extend is unclear.
>
> "the need for performance of the gradient expression supersedes the need for unevaluated gradient expressions"
>
> The argument is that having the unevaluated symbolic gradient (not the differentiated procedural code) helps the performance to grow with human effort. Please see the general comment on evaluation.
>
> "It is entirely unclear to the reviewer whether the prototype implementation of the work is ever used throughout the paper? Would the authors be able to provide evidence that their prototype is actually functional?"
>
> The code is submitted with the paper as supplemental materials. One can download it and run it with Julia 1.10 (julia 1.12 breaks the submitted code).

---

### Author Response · Authors · 2025-11-21
**General comment on evaluation and toolchain**

We thank all reviewers for their thoughtful feedback. We will address a few common concerns here as general comment.

# Evaluation

We would like to emphasize that an important aspect of AD system is not their
absolute performance, but how it grows with human effort. Without controlling
for human effort, any performance advantage of an AD system can be trivially
obfuscated by adding custom primitives and vjp rules to other systems, as
suggested by two reviews for the HF example. Quantifying the human effort to
identify such custom rules is not particularly feasible, so performance
evaluations are potentially misleading. We believe it is better to evaluate
potential advantage through qualitative arguments and representative cases,
which we provided and will further explain.

In the high effort limit, AD systems have to compete with hand-optimized
gradients, and the question is whether using AD can provide a faster path (than
implementation by hand) to near-optimal performance. The answer  based on
[case studies](https://link.springer.com/book/10.1007/978-1-4613-0075-5) appears
inconclusive.
Our work tries to align AD with hand-optimized gradient by providing an
symbolic intermediate expression, so
that doing AD becomes a first step on the path to hand-optimized gradient.

The low effort regime is what motivates the functional DSL, which is designed
to compile symbolic tensor mathematics to numerical primitives and hopefully
save effort for domain scientists. To deliver performance across different
problems and hardware, the compiler needs to optimize over a growing set of
fast primitives described by the same abstract mathematics. What make the DSL
impractical is having to derive, implement, and maintain vjp rules for all
primitives. The B and C rules lift the problem by delaying compilation until
after differentiation, and gradient expressions can be mapped onto primitives
the same way that the model is.

In many cases, the symbolic gradient is desirable in itself for further
symbolic manipulation such as formulating a KTT condition or deriving a
tangent plane projections. It can also be a physical observable such as force $F = \nabla\_r V(r)$,
which reveals how different physical quantities are related.

# Toolchain

The functional DSL is envisioned as a formalization of tensor operations with
programming language concepts for the purpose of leveraging compiler
optimization. The performance is not to be understood as executing a functional
programming language because expressions should be compiled to optimized
library primitives if possible and `for`loops otherwise. For example, `(i, j)
-> sum(k -> A(k, i) * K(k, j))` should compile to a `gemm` call. If BLAS is not
installed for arguments sake, it resorts to a slow triple loop.

```
let result = zeros(size(A, 2), size(K, 2))
    for i in 1:size(A, 2)
        for j in 1:size(K, 2)
            for k in 1:size(A, 1)
                result[i, j] = A[k, i] * K[k, j]
            end
        end
    end
    result
end
```

Mutations can be supported in principle, but it is not productive in our
context. For example, `a[i] += 1` is confluent with `a = (j) -> delta(i, j, 1) + a(j)`, which is pure, so we can differentiate it and optimize the vjp back to a mutation. The problem is that mutations are not abstract
mathematics, which our DSL is designed to describe. Instead, we let the
compiler generate mutating code if no better primitives are available. The main
concern is that the memory IO of generated code is unoptimized, but
differentiating mutation also disrupts the IO pattern, so we choose to avoid
low-level mutations.

# Claim on Partial Evaluation

The authors did not make the claim that all differentiation systems rely on
numerical partial evaluation, and we will rephrase that point. The claim was
empathetically predicated on a specific subtype of problems $\mathcal{P}(w \mapsto f(w)(g))$, which not all
differentiation systems address. The claim was further restricted to standard
techniques and the meaning of partial evaluation was broadened to inlining,
which is one approach to the issue in source transformation systems, as
pointed out by a reference suggested by reviewer zHBj, which we will include.

---

### Meta-Review · Area_Chair_xkoo · 2025-12-13

**Summary:**

Summary of reviewers' assessment:

Strengths:
* The reviewers praise clarity of the paper and careful derivations
* Reviewers point out that the differentiation around a complete combinator basis (B/C) is a valid alternative to the standard automatic differentiation (AD).

Weaknesses:
* Lack of comparison against existing AD in JAX, PyTorch, etc. Absence of quantitative benchmarks such as runtime, memory or compilation times.
*  Misguided claims, such as that standard AD are "blackboxes" and that all AD approaches require tracing/partial evaluation.
*  Lack of practical use: the system relies on statically typed, dimensionally fixed programs and does not allow changes.

**Reviewer Concerns:**

Resolved concerns:
* Addressing the "Blackbox" and Partial Evaluation Claims. The authors clarified the claims and specified that they are targeting a specific subtype of problems.

Unaddressed concerns:
* Absence of numerical evidence and benchmarks on large-scale problems. The reviewers were unconvinced by the author’s argument that the primary value is minimizing human effort.
* Lack of practical usecase
* The proof that the new differentiation rule is provably complete for all differentiable compositions. The authors mentioned they do not have proof.

**Reviewer Scores:**

The main concern of the reviewers remain unaddressed, therefore, the reviewer's scores remain unchanged.

---

### Decision · Program_Chairs · 2026-01-26

Reject